# Novel Therapies in Glioblastoma Treatment: Review of Glioblastoma; Current Treatment Options; and Novel Oncolytic Viral Therapies

**DOI:** 10.3390/medsci12010001

**Published:** 2023-12-23

**Authors:** Siddharth Shah

**Affiliations:** Department of Neurosurgery, University of Florida, Gainesville, FL 32608, USA; siddharth.dr99@gmail.com; Tel.: +91-7710055334

**Keywords:** glioblastoma, malignant brain tumor, neurosurgery, treatment of glioblastoma, oncolytic viruses for treatment of glioblastoma, nanoparticles, viruses

## Abstract

One of the most prevalent primary malignant brain tumors is glioblastoma (GB). About 6 incidents per 100,000 people are reported annually. Most frequently, these tumors are linked to a poor prognosis and poor quality of life. There has been little advancement in the treatment of GB. In recent years, some innovative medicines have been tested for the treatment of newly diagnosed cases of GB and recurrent cases of GB. Surgery, radiotherapy, and alkylating chemotherapy are all common treatments for GB. A few of the potential alternatives include immunotherapy, tumor-treating fields (TTFs), and medications that target specific cellular receptors. To provide new multimodal therapies that focus on the molecular pathways implicated in tumor initiation and progression in GB, novel medications, delivery technologies, and immunotherapy approaches are being researched. Of these, oncolytic viruses (OVs) are among the most recent. Coupling OVs with certain modern treatment approaches may have significant benefits for GB patients. Here, we discuss several OVs and how they work in conjunction with other therapies, as well as virotherapy for GB. The study was based on the PRISMA guidelines. Systematic retrieval of information was performed on PubMed. A total of 307 articles were found in a search on oncolytic viral therapies for glioblastoma. Out of these 83 articles were meta-analyses, randomized controlled trials, reviews, and systematic reviews. A total of 42 articles were from the years 2018 to 2023. Appropriate studies were isolated, and important information from each of them was understood and entered into a database from which the information was used in this article. One of the most prevalent malignant brain tumors is still GB. Significant promise and opportunity exist for oncolytic viruses in the treatment of GB and in boosting immune response. Making the most of OVs in the treatment of GB requires careful consideration and evaluation of a number of its application factors.

## 1. Introduction

In the general population, glioblastoma (GB) is one of the most prevalent malignant brain tumors. Most glioblastomas (around 80–90%) arise de novo and without any prior clinical or histologic signs, especially in elderly individuals. [1]. Currently, one of the most aggressive and incurable cancers is GB [2]. The WHO revised the categorization of malignancies of the central nervous system (CNS) in 2021, incorporating genetic and molecular characteristics into the definition of various glioma subtypes alongside histological ones [3]. Less than 5% of people with GB survive more than five years after diagnosis, with the median overall survival for those with the disease falling between 15 and 20 months [4,5]. Radiation therapy and chemotherapy are usually used after a surgical resection for GB [6]. GB can be treated with temozolomide, bevacizumab, lomustine, intravenous carmustine, and carmustine wafer plants, among other chemotherapy medications [7].

Key risk factors for GB include obesity, cytomegalovirus (CMV), high radiation dosages, and a family history of cancer [8,9,10,11]. GB has been described as having considerable intratumoral heterogeneity, tumor-induced immunosuppression of the microenvironment, and minimal infiltrating immunity [12,13,14]. The blood–brain barrier (BBB) [15], glioma stem cells (GSC) [16,17], and a low tumor mutational load [18,19] are also present.

Recent developments in molecular pathogenesis have led to a better understanding of the GB microenvironment, including its interactions with the human immune system and its genetic, epigenomic, transcriptomic, and proteomic characterization [20]. Among the most promising therapies for GB and brain tumors are novel therapeutic approaches including oncolytic viral therapy [21].

## 2. Overview Glioblastoma

### 2.1. Introduction of Glioblastoma

Initially, it was believed that GB only came from glial cells, but research now reveals that they may also come from other cell types that have characteristics of neural stem cells. These cells are in various phases of development from stem cells to neurons to glia, with molecular changes in signaling pathways acting largely as a determinant of phenotypic diversity rather than diverse cell types of origin [22].

### 2.2. Molecular Description

More than 600 genes were sequenced from more than 200 human tumor samples as a result of genomic profiling and the Cancer Genome Atlas project by Parsons et al., 2008, which revealed the complex genetic profile of GB and established a set of three core signaling pathways that are frequently activated (namely, the tumor protein p53 (p53) pathway, the receptor tyrosine kinase/Ras/phosphoinositide 3-kinase signaling pathway, and the retinoblastoma pathway) [23]. Epidermal growth factor receptor (EGFR) overexpression, mutations in the PTEN gene, and deletion of chromosome 10q are among the genetic changes common to primary GBM. Chromosome 19q deletion, p53 mutations, and isocitrate dehydrogenase 1 (IDH1) mutations are typically observed in secondary GB [24,25,26]. Imaging genomics is a young field of study that investigates relationships between molecular profiles and radiologic characteristics with the potential to one day provide a non-invasive method for identifying, predicting, and correlating genetic variations [27].

Transcriptome studies have become significant methods for grouping cancers into molecular subgroups that differ in terms of their clinical behavior and reaction to treatment [28]. However, when taking into account the extremely aggressive isocitrate dehydrogenase (IDH) wild-type group, the transcriptome categorization has not been able to predict prognosis and pharmacologic vulnerability for specific cancers, such as GBM [29,30]. Specifically, the absence of correlation between survival and physiologically defined subgroups of IDH wild-type GBM has impeded the quest to identify the distinct mechanisms that maintain tumor development in patient subgroups. The transcriptome subgroups utilized to define GBM are preferentially concentrated in tumor cells displaying unique lineage-specific biological states, according to recent results in single cells. The possibility of using the basic biological processes of individual GBM cells to create a clinically meaningful categorization of bulk tumors is still unproven. We reviewed the developed computational approach to extract the core tumor-cell-intrinsic biological states of individual GBM cells from GBM single-cell RNA-sequencing (scRNA-seq) data [31,32,33] and bulk tumors because pathway-based classifications of transcriptomic cancer data have demonstrated higher stability of biological activities and better performance than gene-based classifiers [34]. A novel categorization for GBM was produced by the analysis, which converged on four stable cellular states that include developmental (neuronal and proliferative/progenitor) and metabolic (mitochondrial and glycolytic/plurimetabolic) features. The mitochondrial subtype classifies individuals with better clinical outcomes and is dependent on oxidative phosphorylation (OXPHOS). The mitochondrial group of GBM differs from the glycolytic/plurimetabolic subgroup, which has a poor prognosis and is maintained by the simultaneous activation of numerous energy-producing programs that provide protection against oxidative stress and metabolic variety. This was discovered by multiomics analysis. Targeted metabolic therapy may be beneficial for some GBM patients, as demonstrated by the distinct susceptibility of the mitochondrial subgroup to inhibitors of mitochondrial metabolism.

### 2.3. Risk Factors

One of the few recognized risk factors that has been proven to increase the likelihood of developing gliomas is exposure to ionizing radiation [35]. Radiation-induced GB is generally diagnosed years after receiving therapeutic radiation that was prescribed for another tumor or illness. [36]. Other environmental risk factors for the growth of gliomas include exposure to vinyl chloride, pesticides, smoking, petroleum refining, and the production of synthetic rubber. It has not been demonstrated that exposure to electromagnetic fields, formaldehyde, or nonionizing radiation from cell phones causes GB [37]. Less than 1% of people with glioma have a recognized hereditary disease; however, some specific genetic diseases, such as neurofibromatosis 1 and 2, tuberous sclerosis, Li–Fraumeni syndrome, retinoblastoma, and Turcot syndrome, are associated with an elevated risk of glioma development [38].

### 2.4. Clinical Presentation and Imaging

The anatomical components of the affected brain and the size and location of the tumor can all have a significant impact on how a patient with newly diagnosed GB presents [39,40]. Patients frequently have headaches and localized or progressive neurologic impairments as signs of elevated intracranial pressure. Up to 25% of patients have a seizure as their first symptom, and up to 50% of patients can have one later on in the course of the disease [41,42]. Antiepileptic medicines (AEDs) are already part of the standard of care for patients who present with seizures, although routine use of AEDs in individuals without seizures is not advised [43,44]. At the time of diagnosis, corticosteroids are frequently prescribed to patients to assist in reducing vasogenic edema and relieve related signs and symptoms.

A computed tomography (CT) or magnetic resonance imaging (MRI) scan could be used as part of the initial diagnostic imaging process. Nearly all GB are seen on gadolinium contrast-enhanced MRI, which reveals an irregularly shaped mass with a dense ring of enhancement and a hypointense necrosis center [45].

### 2.5. Current Treatment Options

The current standard of care is for concomitant radiotherapy with temozolomide and the greatest amount of safe surgical resection possible [46]. Because these tumors are commonly invasive and typically located in expressive regions of the brain, such as regions that regulate speech, motor function, and the senses, extensive and full surgical resection of GBM is challenging. Radical removal of the initial tumor mass is not curative due to the high degree of invasiveness, and infiltrating tumor cells typically stay in the nearby brain, causing the disease to develop or return in the future [47].

When feasible, aggressive surgical resection has been shown to be important, and there are trends toward better outcomes in individuals who have undergone more extensive resection [48,49]. Numerous studies [50,51,52,53] have found statistically significant correlations between larger resection depth and longer progression-free survival (PFS) and overall survival (OS). More extensive resections can now be accomplished while maintaining function and quality of life because of advancements in surgical and preoperative mapping procedures [54,55]. Table 1 describes the treatments for new-onset glioblastoma and recurrent glioblastoma.

### 2.6. Role of Immunosuppressive Mechanism in Glioblastoma and Resistance to Immunotherapy

One typical feature of GBM that limits a favorable prognosis is recurrence. Not all patients had access to second-line therapy at this time (about 50% did not receive any treatment while their condition progressed) [56,57]. Numerous studies demonstrate that GBM is associated with an immunosuppressive microenvironment as a result of an increase in factors generated by tumor cells, including FASL, PD-1, indolamine 2, 3dioxygenase (IDO), and STAT3. Additionally, microglia cells have the ability to create IL-1 and TGF-B, which in turn regulate local myeloid and lymphatic immune cells and support systemic immunosuppression [58]. By modifying the expression of several extracellular and intracellular mediators, myeloid cells promote the tumor by ensuring an immunosuppressive microenvironment [59]. These variables all alter the cytotoxic T lymphocyte (CTL) phenotype, which raises the quantities of immunosuppressive markers like PD-1. Several research aims to boost anticancer immune responses by utilizing these approaches. For example, vaccination therapy or anti-PD-1 and anti-CTLA-4 treatments are used to kill tumor cells that have GBM-associated antigens like EGFRvIII [60]. On the other hand, a treatment called viral oncolytic therapy applies a virus that can stimulate the immune system against the tumor. Attenuated oncolytic viruses propagate into tumor cells by taking advantage of the absence of a viral defense system [61].

Although they induce inflammation, elevated intracranial pressure, and CNS neurotoxicity, CAR T lymphocytes—modified chimeric antigen receptor T cells—offer an additional experimental strategy to elicit the anticancer immune response. As a result, this treatment approach is extremely constrained and intricate [62,63,64].

The poor immunogenicity of GBM and the abundance of immunosuppressive stresses in the microenvironment are the causes of resistance to immunotherapies [65].

In recent times, the Hippo pathway has been extensively researched as a molecular mechanism to regulate angiogenesis, invasion, migration, and proliferation of tumors. Numerous investigations demonstrate that YAP may establish contact between immune cells and tumors, especially with TAMs [66]. Indeed, the recruitment and activation of many inflammatory cytokines, such as IL-6, which regulate the tumor immune response and tumor development, is made possible by the presence of YAP in the nucleus. Moreover, TAMs seen in gliomas generate and secrete IL-6, which has the ability to stimulate the growth of glioma stem cells and trigger the build-up of TAMs in a feed-forward loop.

Consequently, the discovery of the Hippo pathway molecular target and viral oncolytic therapeutics might lessen GBM’s immunosuppressive and chemoresistance characteristics, making the molecular route of interest for research purposes.

## 3. Novel Oncolytic Viral Therapy for Treatment of Glioblastoma

OVs are useful in treating GB because of their ability to replicate virally quickly in rapidly proliferating cells, their absence of distant metastases, and their alignment with the brain environment [67,68]. The anticancer immune response begins by converting “cold tumors” that are immunosuppressed by the microenvironment into “hot tumors” that are sensitive to the immune system [69,70,71,72]. Inducing an immunological response to inadvertently kill cancer cells through several mechanisms, including apoptosis, necrosis, and autophagy, is known as immunogenic cell death (ICD) [73,74,75]. Releases of damage-associated molecular patterns (DAMPs), viral pathogen-associated molecular patterns (PAMPs), tumor-associated antigens (TAAs), and several other cytokines are indicative of this [76,77]. Oncolytic viruses enhance the function of antigen-presenting cells (APCs), which reach lymph nodes to recruit cytotoxic CD8+ T lymphocytes (CTLs) and attract them to the infection site, where they destroy tumor-inducing cells [78,79,80]. Figure 1 depicts the molecular process mentioned.

Virotherapy is currently thought to be a promising immunotherapy for GB. The two types of OVs are replication-competent OVs, which only reproduce in cancer cells, and replication-deficient viral vectors, which are employed as carriers for additional therapeutic genes. Viruses that are produced through genetic engineering and naturally occurring viruses make up the first group [81,82]. The first group includes Newcastle disease viruses (NDV), reoviruses, and parvoviruses. The genetic modification of adenoviruses (Ad), herpes simplex virus (HSV), vaccine viruses (VV), vesicular stomatitis viruses (VSV), polioviruses, and measles viruses (MV) can decrease their pathogenicity and increase their tumor selectivity. To facilitate virus propagation and subsequent stimulation of the antitumor immune response, certain OVs use specialized receptors expressed on tumor cells.

In clinical studies for the treatment of GB, more than 20 oncolytic viruses have been examined. HSV-1 [83,84,85], adenovirus [86], reovirus [87], MVs [88,89], NDVs [90], and poliovirus [91] are a few of them. Novel techniques for OV distribution are being developed to overcome the BBB limitation. One such technique is the convection-enhanced distribution (CED) of the recombinant nonpathogenic polio-rhinovirus chimera (PVSRIPO) [92]. CED is a cutting-edge novel technology that transfers therapeutic chemicals in the interstitial regions of the CNS using a pressure gradient in a catheter [92]. For virotherapy to be successful, oncolytic viruses must be delivered effectively and safely. Intratumoral delivery was selected as the main method due to the difficulty of delivering viruses to the CNS and the immune system’s ability to eliminate them [93]. However, it is preferable for the oncolytic virus to be administered consistently to both primary and metastatic tumor locations [94]. Thankfully, glioblastoma seldom spreads metastatically outside of the central nervous system [95]. The oncolytic viral studies and their results are described in Table 2.

### 3.1. DNA Viruses

#### 3.1.1. Herpes Simplex Virus Type I

The HSV-based oncolytic virus was the first modified viral strain assessed for experimental treatment in a murine GB model [97]. A double-stranded DNA virus belonging to the Herpesviridae family, HSV-1 has double-stranded DNA. Nectin-1, a cell surface protein that is widely expressed in GB, serves as a binding site for HSV-1 [98]. The US Food and Drug Administration (FDA) approved talimogene laherparepvec (T-VEC) in 2015 to treat advanced, unresectable melanoma that is not amenable to surgery [98,99]. T-VEC expresses granulocyte-macrophage colony-stimulating factor (GM-CSF). GB patients are participating in clinical studies with many modified HSV constructs, including G207, HSV-1716, M032, and MVR-C252. Tumor selectivity was improved in all HSV recombinants by removing both copies of the RL1 gene, which codes for the viral protein (ICP34.5 producing neurovirulence) [100,101]. HSV G207, a recombinant strain with a malfunctioning viral ribonucleotide reductase (RR), was unable to reproduce in healthy cells. By producing the homologic gene, tumor cells make up for the loss of RR [102]. Third-generation oncolytic HSV-1 strain G47, sometimes referred to as DELYTACT, has demonstrated encouraging results in phase II clinical studies [103]. G47 was created by modifying the ICP47 gene with a deletion to improve the immune system’s ability to recognize tumors utilizing MHC class I [103]. For 19 adult patients with glioblastoma that was either residual or recurrent, a phase II single-arm study using G47Δ was initiated. Up to six intratumoral doses of G47Δ were given. A one-year survival rate of 84.2% indicates that the stated endpoint was achieved. The most frequent side effect was fever in 17 out of 19 patients. The patients’ biopsies revealed CD4+/CD8+ cells that had infiltrated the tumors [104]. A second phase I/II study of G47 in patients with recurrent or progressive glioblastoma found a median overall survival of 7.3 months and a 38.5% one-year survival rate [105]. This research led to the conditional approval of G47Δ in Japan in 2021. Tumor-specific promotors (such as Nestin-1) are used in the recombinant virus rQNestin34.5 to control ICP34.5 expression for enhanced cytolytic activity in tumor cells [106]. M032 is a recombinant HSV that produces human interleukin 12 (IL-12) to increase interferon-gamma (IFN-) production and anticancer activities [107]. The ICP6 and ICP34.5 genes have been deleted in NG34, a novel variant of rQNestin34.5 produced by the oncolytic HSV (oHSV). Although less hazardous in vivo than its predecessor, NG34 demonstrated comparable efficacy [108]. Another oHSV, rRp450, has an insertion of CYP2B1 and a deletion of ICP6, making it possible to activate the prodrug cyclophosphamide (CP) [109]. This virus boosted effectiveness and survival in tumor-bearing mice once the CP agent was introduced. OV-CDH1 is a modified herpes simplex virus (HSV) that produces E-cadherin to improve viral proliferation in tumors by increasing the oncolytic impact and reducing NK-mediated immunity in the infected cells [110,111]. By producing matrix metalloproteinase 9 (MMP), which targets the EGFRvIII mutant antigen specific to the tumor, another oHSV promotes the tumor’s spread [112]. Furthermore, this virus possesses a unique recognition site for the miR-124 miRNA, which enhances the miRNA’s selectivity for tumor brain cells and suppresses the vital viral protein ICP4 in healthy glial cells [113]. The Flt3L-expressing oHSV demonstrated complete GB clearance in preclinical studies [114]. Another oHSV that produces TRAIL, a protein that triggers TNF-CD95L and encourages apoptosis, exhibits a cytotoxic impact in GB models in mice with prolonged survival rates [115]. It was possible to reverse the effects of oHSV treatment with PD-1 antibody against GB mice models [116]. Whether administered intravenously or intratumorally, there are currently several flaws in the use of ohsv that restrict its therapeutic impact. Although it is difficult for virus particles to move to the lesion region outside of the injection place, intramoral injection can guarantee that they reach the lesion directly. The virus may infect all cancer cells by intravenous injection, which is very useful for treating metastatic lesions.

#### 3.1.2. Adenovirus

Adenoviruses are non-enveloped, double-stranded DNA viruses that have an icosahedral shape [117]. Conditionally replicative adenoviruses (CRads) have been developed in a number of variations and have demonstrated promising anti-GB activity in clinical settings [118]. Furthermore, individuals with GB alone or in conjunction with other ICIs are now undergoing clinical studies with a genetically engineered adenovirus (Table 1). Reportedly, malignant gliomas have been effectively treated with an alternative gene-mediated cytotoxic treatment approach [119]. The phase II clinical study employed adenovirus glatimagene besadenovec (AdV-tk), which possesses the HSV thymidine kinase gene and kills cancer cells upon interacting with alacyclovir [120]. Removing the viral replication genes is one way to stop off-targets in normal cells, which can still proliferate in tumor cells. China has approved H101 (Oncorine), which is similar to oncolytic adenovirus ONYX-015, for the treatment of head and neck cancer [121,122]. A loss in the E1B-55K gene limited the recombinant adenovirus ONYX-015 oncolytic’s capacity to replicate to tumors with p53 abnormalities [123]. Two genetic changes were found in DNX-2401, an OV based on serotype 5 Ad (Ad5) [124]. When the E1A gene is removed and an RGD-4C motif is added to the fiber’s HI loop, the virus switches to replicate in cells with impaired pRB pathways that generate v3- and v5-integrins, both of which are indicators of glioma cells [125]. Due to this alteration, adenoviruses may infiltrate cells even in the presence of trace levels of their major receptor, the cox-sackie-adenovirus receptor, on brain tumor cells [126]. The OX40L gene is produced by the second generation of DNX-2401, DNX-2440 (also known as Delta-24-RGDOX), to improve T-cell-mediated immunity by encouraging the proliferation of CD8+ specific-tumor T cells [127,128]. To further stimulate T cells, the oncolytic adenovirus Delta-24-ACT produces the 4-1BB ligand in animal glioma models [129]. With the help of glucocorticoid-induced TNFR-family-related gene ligand (GITRL), another oncolytic adenovirus called Delta-24-RGD was able to increase mice survival and stop further exposure to glioma cells [130]. Furthermore, utilizing an alternative oncolytic adenovirus that expresses the co-stimulator OX40 ligand (OX40L) resulted in the activation of cancer-specific immunity in vivo as well as the expansion of CD8+ T cells [131]. Although it is still in its early stages, the clinical usage of treatments based on oncolytic adenoviruses has shown great promise. It is particularly appealing in the therapeutic setting due to its broad compatibility with currently approved medicines (without increasing toxicity), its various cell-killing activities, and its ability to induce immunogenic cell death.

#### 3.1.3. Parvoviruses

The Parvoviridae family of single-stranded icosahedral DNA viruses includes parvoviruses. Various animal species can be infected by one of about 134 different parvovirus serotypes [132]. A small oncolytic virus called H-1 parvovirus has shown anticancer efficacy against GB [133]. Additionally, H-1PV causes glioma cells to undergo apoptosis and breaks down their resistance to a number of chemotherapeutic drugs [134]. Human U87-MG glioma models in rats showed tumor shrinkage in preclinical studies with the H-1PV [135]. As a result, the ParvOryx01 trial for individuals with recurrent GB (NCT01301430) was started. ParvOryx01 proposed that tumor-infiltrating lymphocytes (TILs) were responsible for inducing immune responses in the removed tumor tissues of GB patients [136]. In high-grade human gliomas, radiation promotes H-1PV viral oncolysis, which may be considered in animal glioma models [119]. Bevacizumab with H-1PV improved the mean survival to 15.4 months in five patients with recurrent GB and produced remission in three of them [137]. These results are associated with the synergistic effect of bevacizumab and H-1PV in controlling GB TME and decreasing VEGF [138]. The first clinical evidence of using H-1PV in combination with bevacizumab and an immune checkpoint inhibitor (nivolumab) was shown in a multimodal clinical study including three patients with recurrent GB. Every participant achieved clinical improvement and confirmed tumor shrinkage, with 78% of cases showing complete or partial remission [139]. All of these results suggest that parvoviruses may be useful in immunotherapies against GB. New clinical trials have confirmed PVs’ safety and tolerability and offered a proxy for confirmation of therapy effectiveness for cancer. Despite the hopeful nature of these data, patient responses following treatment with H-1PV did not match the remarkable outcomes shown in preclinical assessment. Consequently, oncolytic PV treatment still has to be optimized and developed further. Like any medication, oncolytic PVs need a suitable economic climate for clinical study, which is now a significant roadblock in their development.

#### 3.1.4. Myxoma Virus

A part of the poxvirus family with double-stranded DNA is the myxoma virus (MYXV) [140,141]. MYXV can cause an oncolytic impact when it replicates in cells like GB that lack an interferon system [142]. The deletion of the viral antiapoptotic protein M011L in the M011L-deficient MYXV virus boosted apoptosis in malignant glioma cells [143]. A prospective candidate OV that has shown promise in several preclinical cancer models is MYXV. Furthermore, MYXV is an appealing OV platform due to its remarkable safety profile outside of rabbits, its extremely selective tropism for a wide variety of cancer cell types, and the limitation of viral multiplication in original non-transformed human cells.

#### 3.1.5. Vaccinia Virus (VV)

The Poxviridae family includes the double-stranded DNA virus known as a vaccine. Smallpox was eradicated with the aid of VV. Because VV may infect any kind of cell by membrane fusion with a non-integrative replication cycle, it is a suitable platform for oncolytic viral engineering against GB [144]. The only recombinant VV that has shown therapeutic benefits in brain tumor patients is TG6002 [145]. The TG6002 genome contains two additional gene deletions for the RR and thymidine kinase (TK) genes. By introducing the FCU1 gene, the chemotherapy prodrug 5-flucytosine (5-FC) was also converted into 5-fluorouracil (5-FU) [146]. Prior research has demonstrated that systemic PD-1 blockade medication and local injection of oncolytic VV together are more effective than either treatment alone. As an alternative, oncolytic VV or HSV that was engineered to express PD-1 blockades demonstrated an anticancer effect comparable to that of using OV and PD-1 blockades together. In this work, the authors showed for the first time that arming VV with a scFv against TIGIT dramatically improved the parental VV’s antitumor effectiveness by altering the TME’s immunological state. For the first time, it was shown that the antitumor effectiveness of VV equipped with the scFv against TIGIT was further increased by the additional combination of PD-1 or LAG-3 inhibition.

### 3.2. RNA Viruses

#### 3.2.1. Measles Virus

The measles virus (MV) is a single-stranded RNA virus with a negative sense that belongs to the Paramyxoviridae family [147]. The MV enters cells via engaging with the overexpressed CD46 cell receptor on tumor cells as well as the viral hemagglutinin (H) protein [148]. Recombinant MVs entered clinical trials after glioma xenografts showed strong anticancer efficacy in them [149,150]. To monitor viral expression in cells, such recombinants express the human sodium iodide symporter (NIS) or the human carcinoembryonic antigen (CEA) [151]. NIS allows for the monitoring of viruses using various isotopes and has the potential to enhance viral cytopathogenic effects [152,153]. The present studies deliver the maximum practicable dosages as a result of the observed dose–response correlations. Even with the introduction of technologies like synthesis in serum-free cell culture, tangential flow filtration, and diafiltration, the necessary high-titer, highly purified good manufacturing practice (GMP)-grade recombinant MV remains difficult to produce on a wide scale. However, considering the diversity of MV as a platform for oncolytic vectors, the high safety record of MV vaccinations, the genetic stability and biosafety profile of recombinant MV, and the biosafety profile of MV, these efforts appear to be justified.

#### 3.2.2. Vesicular Stomatitis Virus (VSV)

The VSV is a single-stranded, negative-sense RNA virus that belongs to the Rhabdoviridae family. The spike glycoprotein (G) of the VSV is linked to the low-density lipoprotein receptor (LDL-R), a cell receptor that is broadly dispersed [137]. The VSV is employed as an oncolytic drug against various malignancies by replicating in tumor cells via the abnormalities in their interferon system [138,139]. rVSV (GP) and VSV-EBOV are terms for engineered VSVs that have had the envelope glycoprotein (GP) substituted with GP from the Ebola virus and the non-neurotropic lymphocytic choriomeningitis virus, respectively [140,141]. Despite entering a phase I clinical trial, an oncolytic VSV is suppressed by viral-mediated production of interferon (IFN)β, which has been demonstrated to increase the virus’s safety. Other methods to increase the safety of VSV include changing the tropism by pseudotyping with a heterologous virus’s glycoprotein. In light of this, it has been demonstrated that rVSV-GP, a pseudotyped vaccine vector containing the glycoprotein of the lymphocytic choriomeningitis virus, is both safe and effective.

#### 3.2.3. Reoviruses

When the Ras-signaling pathway is triggered in glioma cells, reoviruses—double-stranded RNA non-enveloped viruses—can multiply in the cells [139]. Reovirus RNA genome mutations occur quite quickly. This offers a degree of flexibility that may be used to choose reovirus variants with higher levels of oncolytic activity. The reovirus genome may also be genetically altered, providing further possibilities for boosting the oncolytic activity. One such method is the insertion of tiny therapeutic transgenes [139]. Reoviruses having the benefit of not being linked to any severe human diseases, and there are now more and more clinical trials including the use of reovirotherapy to treat cancer. With the modest effectiveness of reovirus as a monotherapy, the emphasis has shifted to combination regimens thus far. Apart from genetic alteration, conventional bioselection is an additional mechanism that may be employed to augment the oncolytic capabilities of reoviruses. Recognizing that reoviruses are not oncolytic agents is a positive thing. Thus, the reovirus’s capacity for environmental adaptation may aid in the selection of more potent forms. The jin reoviruses are one type of such mutation. The wild-type reovirus was driven to evolve a way around the JAM-A reliance due to its extended proliferation on cells lacking JAM-A on their surface [139]. This idea can be investigated for tumor forms that are resistant to infection by reoviruses at other phases of the viral replication cycle, including cell lysis or endosomal escape. Certain malignancies have developed defense mechanisms against cancer treatments that depend on inducing apoptosis and avoiding signaling pathways involved in cell death.

#### 3.2.4. Newcastle Disease Virus (NDV)

The Paramyxoviridae family includes the negative-sense, single-stranded RNA-enveloped NDV virus [137]. Interferon-stimulated genes (ISGs) are expressed by the NDV, which is largely an avian virus that preferentially replicates in tumor cells and triggers the type I interferon response in humans [148,149]. Studies suggest that NDV may be useful against GB [150]. The highly contagious avian disease NDV is a member of the Paramyxoviridae family of viruses. It results in a sickness that causes large financial losses and for which the World Organization for Animal Health (OIE) must be notified in writing. NDV is also recognized as an oncolytic virus, able to cause organelle-specific autophagy in the Golgi apparatus, peroxisomes, and mitochondria while also replicating specifically in tumor cells. NDV is a useful model for future cell biology studies because of this. The authors postulated that SIRT3 can function as a crucial gatekeeper for the balance between glycolytic and mitochondrial oxidative metabolism in the control of energy production for viral replication because of its critical involvement in cell metabolism [150].

#### 3.2.5. Seneca Valley Virus Isolate 001 (SVV-001)

A member of the Picornaviridae family of positive-sense single-stranded RNA, the SVV-001 [151] has shown oncolytic activity against solid tumors, with a particular affinity for cells expressing the endothelium receptor TEM8/ANTXR1 [152]. The transmembrane glycoprotein adhesion molecule TEM8/ANTXR1 is more prevalent in some cancer types and mediates cell motility and its interactions with the extracellular matrix (ECM) [153]. TEM8/ANTXR1 is the first biomarker for SVV-based oncolytic viral treatment [154]. SVV-001 given intravenously has anticancer properties and is able to pass the BBB [155]. As the first oncolytic virus ever tested in children globally, SVV-001 is also the first to be examined in a phase I study for recurrent/refractory cancers in both adults and pediatrics. The use of this virus in tumors showing neuroendocrine traits is supported by favorable preclinical evidence in SCID mice; however, objective clinical responses were absent in the phase I studies conducted in adults and children. Considering how powerful this virus was when it first emerged in SCID mice models, completely eliminating tumors with a single SVV-001 injection, this is a little unexpected. Given that these mice were Rag2 SCID models, they were severely immunocompromised and hence unable to establish an immune response involving T-regulatory cells and/or create neutralizing antibodies against SVV-001, as shown in human investigations. This may have contributed to some of the observed effects. Nevertheless, without the need to determine the highest dosage that may be tolerated, systemic delivery of the virus did show promise for safety in both pediatric and adult populations.

#### 3.2.6. Polioviruses

The Picornaviridae family of positive-sense single-strand RNA viruses includes polioviruses [156]. The CD155/PVR receptor, which is typically overexpressed on cancerous cells, is used by polioviruses to infect cells [156].

The internal ribosome entry site (IRES) of an attenuated poliovirus type 1 (Sabin) vaccination strain was substituted with an IRES from a human rhinovirus type 2 in order to reduce the potential neurovirulence [157,158]. In the phase I study (NCT01491893) looking at the intratumoral CED of PVSSRIPO in patients with recurrent GB, the safety and absence of neurovirulence were established. Consequently, the PVSRIPO was granted a breakthrough therapeutic classification by the FDA in 2016 [159]. Additionally, the figures demonstrate that the trial’s survival rate was higher than that of the historical controls, with rates at 24 and 36 months rising by 21%. We anxiously await the results of a phase II study (NCT02986178) investigating PVSRIPO alone or in combination with lomustine in patients with GB [159].

Human rhinovirus type 30 has been inserted into the IRES of the novel recombinant poliovirus type 3 vaccination strain RVP3, which replicates only in tumor cells and leaves healthy cell lines unaffected [160]. On primary glioma cells from various patients as well as various glioma models, RVP3 demonstrated oncolytic efficacy [161].

A study carried out in this patient group, assessing intratumoral convection-enhanced transport of the recombinant nonpathogenic polio-rhinovirus chimera (PVSRIPO), found that all 61 patients who got PVSRIPO had a median overall survival of 12.5 months, longer than the historical control group’s 11.3 months [162].

#### 3.2.7. Sindbis Virus

The Togaviridae family of positive-sense single-stranded RNA viruses includes the Sindbis virus [163]. Via binding to the laminin receptor (LAMR), Sindbis infects cancer cells and causes death in glioma cells [164,165] via tyrosine phosphorylating protein kinase C delta. The Semliki forest virus (SFV4miRT) contains target sequences for miR124, miR125, and miR134 inserted into it; it is expressed more in healthy CNS cells than in glioma cells [166]. As a result, this virus has a decreased neurotropism, oncolytic effectiveness, and safer profile [167,168]. According to recently published research [169,170], the Zika virus (ZIKV) can infect GB stem cells (GSCs) and exhibit oncolytic activity on them.

This suggests that modifying ZIKV to more precisely target GB in the absence of normal neuronal cells may enhance treatment results [171,172]. ZIKV-LAV was created by a 10-nucleotide deletion in the 3′ untranslated region (3-UTR) of the genome. It exhibits lower neurovirulence and higher GB oncolytic activity [173,174].

Clinical studies for GB have typically demonstrated that oncolytic viruses are safe and efficient against glioma cells, albeit few of these trials have progressed to phase III. Sitimagene ceradenovec, an adenoviral vector expressing the HSV thymidine kinase gene, was studied in the phase III clinical study “ASPECT” in combination with intravenous ganciclovir. However, there was no appreciable effect on overall survival [175]. A phase III trial for Toca511, a retroviral vector containing the gene for cytosine deaminase (CD), has begun. Toca511’s CD gene transforms the cancer-killing compound 5-flucytosine into 5-fluracil [176,177]. This trial was also stopped for undisclosed reasons. It is crucial to remember that while OVs have been shown to be safe and beneficial in preclinical research, clinical efficacy has not yet attained the intended degree [178,179]. Table 3 discusses all the oncolytic viruses mentioned in this section along with their advantages.

## 4. Discussion

One of the worst and most lethal diseases is still GB. The use of oncolytic viruses to treat and stimulate the immune system against GB has considerable promise. A number of issues need to be resolved in order to maximize the benefits of OVs. Considerations include the OV’s ability to multiply and infect tumor cells, the condition of the tumor microenvironment, the degree of immune cell infiltration, and the possibility of inducing an anticancer response. This study reviewed the present state of GB therapy alternatives, including oncolytic viruses and nanoparticles that are currently being utilized or studied in clinical studies. Virotherapy is regarded to be a promising immunotherapy for GB at the moment. There are two types of viral vectors (OVs): replication-competent OVs, which only multiply in cancer cells, and replication-deficient OVs, which are utilized as carriers for additional therapeutic genes. The first group consists of viruses created through genetic engineering and naturally existing viruses. Newcastle disease viruses (NDV), reoviruses, and parvoviruses comprise the first category. To increase tumor selectivity and decrease pathogenicity, genetic modifications can be made to adenoviruses (Ad), herpes simplex virus (HSV), vaccine viruses (VV), vesicular stomatitis viruses (VSV), polioviruses, and measles viruses (MV). Some OVs use specific receptors that are expressed on tumor cells to help spread the virus and then trigger the anti-tumor immune response.

Many OVs are now in clinical studies, and numerous factors, including viral delivery and appropriate dosage, are being investigated. Ex vivo customized models might be constructed to choose the best oncolytic virus for each patient. Several OVs will be evaluated in future clinical studies to see if they connect with tumor heterogeneity and immune state complexity. As new techniques for virotherapy are attempted, new monitoring and evaluation requirements should be implemented. Prospective research endeavors aimed at clarifying the process of OVs in various GB models ought to employ inventive genetic engineering techniques and viral delivery methods. In conclusion, although virotherapy alone may be beneficial, combining immunotherapy and oncolytic viral techniques with customized strategies may lead to a more successful course of treatment for individuals with GB.

## 5. Conclusions

We can safely state that oncolytic viral therapy is a well-established cancer treatment modality at this point. The efficacy of oncolytic virus therapy is anticipated to increase when coupled with immunotherapy since the common feature that plays a significant role in showing anticancer effects during oncolytic activities is the formation of specific antitumor immunity. Functional transgenes would enable oncolytic viruses to be equipped with a wide range of anticancer capabilities in the future. Based on the kind and stage of cancer, a combination of suitable viruses may then be selected from this panel. Oncolytic viral therapy appears to be the beginning of a new age in cancer treatment, where patients have the freedom to choose this treatment option. For the purpose of creating oncolytic viruses, a reiterative feedback loop—in which the outcomes of clinical trials inform and influence the design of succeeding generations of viruses—is preferred over a unidirectional method. Particularly with regard to virotherapy in the brain, preclinical laboratory research can only partially address the unique challenges this field faces. The biological effects of viruses vary greatly depending on the species under investigation, in contrast to small molecule therapies. Human viruses including poliovirus, AdV, and HSV are greatly attenuated in tumor models found in rodents, but they may be less so when administered to people. On the other hand, non-human infections including PRV, SIN, and VSV can be harmful to mice, which makes preclinical survival research extremely difficult. Oncolytic viruses are distinct from conventional medications in a number of ways. Since they are live viruses, their effective dosages may vary depending on how quickly they multiply in a therapeutic setting. Little information is currently known on the relationship between viral dosage, in vivo replicative capability, and treatment response. To create safe and effective dose guidelines, more research on viral replication and clinical response in pertinent preclinical models and clinical trials is necessary. In several clinical studies carried out across a wide spectrum of malignancies, oncolytic viruses have so far been linked to a generally acceptable safety profile. However, given these agents’ capacity for replication, infection control measures—such as proper handling, storage, preparation, and delivery of the virus—must be taken seriously. The actual risk of infection is contingent upon the type of virus, co-occurring medical problems in patients, close household contacts, and healthcare personnel who may come into touch with the virus. Furthermore, a lot of oncolytic viruses have recombinant DNA elements, and therefore there are theoretical worries about the possible effects of these gene segments and the possibility that they would recombine with wild-type viruses in the environment. A new level of safety concerns arises when these non-human viruses enter clinical trials because the effects of environmental contamination and transmission must be taken into account. Oncolytic viruses provide special manufacturing and regulatory challenges since they are live reproducing viruses. Since tissue cultures are used to proliferate most viruses, techniques for producing high-titer viruses, screening for adventitial infections, and evaluating virus purity and replication capacity are necessary. Therefore, it is necessary to take into account protocols for laboratory safety throughout the manufacturing and vialing processes, product validation and purity, and design quality that arises from producing biologics in cell culture. It has been difficult to produce the very high titer lysates needed for therapeutic dosage for some viruses, which can provide a problem for the biotechnology manufacturing industry. These elements have been examined and require more investigation. Oncolytic viruses, however, have been linked to a highly favorable risk-benefit ratio, and thus more research and development of this novel class of medications is expected, with a focus on combination therapies in particular. A novel class of medications is introduced by oncolytic viral immunotherapy, a very promising method for treating cancer patients.

## Figures and Tables

**Figure 1 medsci-12-00001-f001:**
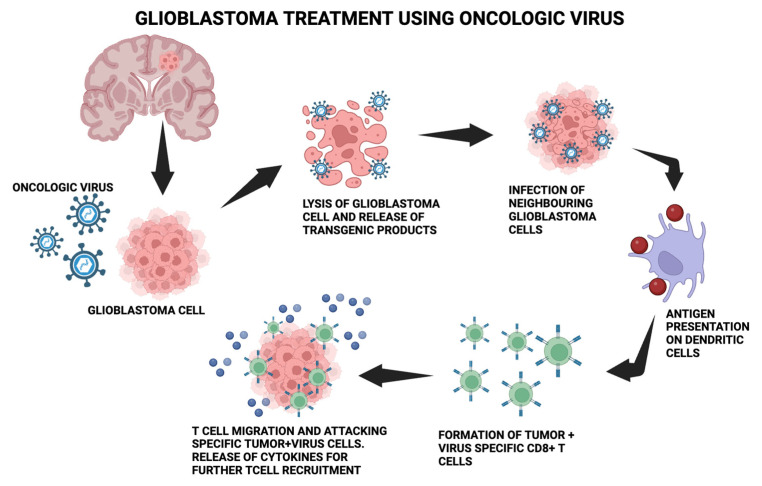
The molecular process behind using the oncologic virus for glioblastoma treatment. The oncolytic virus is introduced into glioblastoma cells, which causes lysis and the release of transgenic products. Neighboring glioblastoma cells get infected, leading to antigen presentation on dendritic cells and further leading to the formation of tumor + virus-specific CD8+T cells. T-cell migration and attacking of specific tumor + virus cells occurs.

**Table 1 medsci-12-00001-t001:** Treatments for new-onset glioblastoma and recurrent glioblastoma according to The American Society of Clinical Oncology (ASCO) and the Society of Neuro-Oncology (SNO).

New Onset Glioblastoma	Recurrent Gliobastoma
Patients with freshly diagnosed glioblastoma, IDH-wildtype, should be offered concomitant TMZ and RT.	Magnetic resonance imaging (MRI) augmented with gadolinium contrast is advised.
Patients who have undergone concurrent RT with TMZ should be offered adjuvant TMZ for six months.	18-FDG is not advised for use in regular diagnostic procedures.
After chemoradiation treatment, individuals may receive adjuvant TMZ in addition to alternating electric field therapy.	Patients with symptomatic pGBM are advised to undergo cytoreductive surgery.
Treatment with bevacizumab is not advised.	It is not recommended to reevaluate the methylation status of 06-methylguanine-DNA methyltransferase (MGMT) and the state of isocitrate dehydrogenase (IDH).
Hypofractionated radiation therapy combined with TMZ is a suitable choice when the projected survival benefits of a six-week radiation treatment combined with TMZ may not exceed the hazards.	The activity of the mismatch repair enzyme (MRE) 1/programmed death ligand (PDL) is not a helpful part of routine diagnostic testing.
When treating patients who are older, in poor performance status, or who have concerns regarding toxicity or prognosis, hypofractionated RT alone or TMZ alone are reasonable options.	Epidermal growth factor receptor (EGFR) amplification may be useful for diagnosis if it has never been tested before.
No therapeutic approach is recommended or discouraged; instead, if at all feasible, these patients should be directed to take part in a clinical study.	Patients who are interested in or eligible for clinical trials or molecularly guided treatment may have their large panel sequencing needs taken into consideration.
	Treatment with TMZ may be beneficial, particularly if it is continued for more than five months after stopping the medication.
	When an aged patient’s MGMT promoter status is methylated, fotemustine is recommended.
	For adult patients, tumor treatment fields (TTFs) with additional chemotherapy may be taken into consideration.

**Table 2 medsci-12-00001-t002:** Oncolytic viruses in clinical trials for GB patients [96].

Oncolytic Virus	Outcomes
HSV-1 MVR-C5252 (C5252) that has been genetically altered	Safety and tolerability dose-limiting toxicities (DLT) and maximum tolerated dose (MTD)
Modified genetically HSV-1 M032	MTD
G207 administered once via catheters into tumors	Safety and tolerability
Oncolytic viral vector rQNestin34.5v.2 HSV	MTD
DNX-2440, a genetically modified adenovirus	Safety, overall survival, and objective response rate
Adenoviral Nsc-crad-s-pk7	-
Adenovirus DNX-2401	MTD and incidence of adverse event
H-1 parvovirus (H-1PV)	Safety and tolerability
PVSRIPO, a recombinant nonpathogenic polio-rhinovirus chimera, is injected into a tumor via CED	MTD, dose-limiting toxicities, recommended phase 2 dose
PVSRIPO	14 days following PVSRIPO therapy, toxicity
PVSRIPO delivered into a tumor via CED	objective radiographic response, duration of objective radiographic response at 24 and 36 months
Live, replication-competent wild-type reovirus REOLYSIN	MTD, DLTs, and six-month response rate
Combination of modified vaccinia virus TG6002 and 5-FC	DLTs and the six-month course of the tumor

**Table 3 medsci-12-00001-t003:** Oncolytic viruses and their advantages.

Oncolytic Virus	Advantages
Herpes simplex virus type I	It is a DNA virus that does not integrate into the host genome and has a large genome. Comparing wild-type HSV-1 to other viruses in development, its pathogenesis is usually milder. The usage of an envelope, which makes it easier for the virus to be retargeted through genetic engineering.
2. Adenovirus	○ It has a high infection efficiency and a big cargo limit. ○ Its transient gene expression does not last long when acting on undesired targets, and therefore safety is improved.
3. Parvoviruses	Has a low inherent capacity for infection and is not as closely linked to human illness. The parvovirus can withstand high temperatures and is stable in pH extremes. Can sustain the inactivation process of gamma ray.
4. Myxoma virus	○ Specifically target and eliminate human cancer cells. ○ Has been tried in many cancer trials and has proven efficacy.
5. Vaccinia virus	Short (8 h) life cycle that is entirely contained in the cytoplasm reduces the possibility of genome integration. The virus may transcribe mRNA without the assistance of the host. Does not possess a unique cell entry surface receptor.
6. Measles virus	○ It has a superior safety profile, is non-genotoxic, and has high immunogenicity. ○ Plenty of engineering opportunities available.
7. Vesicular stomatitis virus	Potent inducer of the infected cells’ apoptosis. Triggers chemotherapy-resistant tumor cells more susceptible to death.
8. Reoviruses	○ Low side effects, a dose that is well tolerated. ○ Stimulate the intrinsic/extrinsic pathway to cause apoptosis.
9. Newcastle disease virus	Evades the issue of the virus’s virulence and preexisting immunity in people. Has potent immunostimulatory qualities. Utilized to combat a range of cancer types.
10. Seneca valley virus	○ Does not incorporate DNA into the human genome. ○ Makes it easier to create recombinant viruses with higher therapeutic indices.
11. Polioviruses	There is real evidence that the poliovirus can target, infect, and destroy cancer cells that are generated from neuroectodermal and ectodermal malignancies. Transgenic variants have good safety.
12. Sindbis virus	○ Blood-borne virus, which enables it to infect the majority of bodily tissues. ○ Easy to engineer. ○ Uses the 67 kDa high-affinity laminin receptor (LAMR), which is commonly overexpressed in tumor cells relative to normal cells, to cause apoptosis.

## Data Availability

Not applicable.

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
