# Peer review of "Novel Therapies in Glioblastoma Treatment: Review of Glioblastoma; Current Treatment Options; and Novel Oncolytic Viral Therapies"

_medsci, 2023, doi:10.3390/medsci12010001_

Round 1

Reviewer 1 Report (Previous Reviewer 2)

Comments and Suggestions for Authors

In the revised form, this manuscript offers a comprehensive and valuable review of novel therapies in GB treatment. It effectively addresses a significant gap in the field and provides a well-structured exploration of the topic. To further enhance its quality, the authors might consider providing additional details regarding the methodology. Overall, the manuscript is a valuable addition to the field and merits publication.

Author Response

Thank you for the review comments.

I have updated the methodology with more information on the process.

". The study was based on the PRISMA guidelines. Systematic retrieval of information was performed on PubMed. 307 articles were found in a search on oncolytic viral therapies for Glioblastoma. Out of these 83 articles were Meta-analyses, Randomized Controlled Trials, Reviews, and Systematic Reviews. 42 articles were from the years 2018 to 2023. Appropriate studies were isolated, and important information from each of them was understood and entered into a database from which the information was used in this article"

Reviewer 2 Report (New Reviewer)

Comments and Suggestions for Authors

In the last ten years, a large number of review articles have been written that are similar in topic to this review, which indicates the relevance of research in this direction. Although the title of the review should reflect new therapeutic approaches to the treatment of glioblastoma, the number of references from the last five years is about 20%. Therefore, I recommend inserting more up-to-date information in some sections and citing sources from the last five years. In terms of style, this review has some similarities with the article, especially in the order of the chapters. It’s strange that the discussion section is smaller than the Conclusions section, it should be the other way around. Conclusions should briefly summarize the information presented.

Overall, I recommend this review for publication after updating it with more up-to-date information and references from the last 5 years.

Author Response

thank you for the review comments.

16 new referenced have been added from 2023 to 2018 as per your suggestions.

Reviewer 3 Report (New Reviewer)

Comments and Suggestions for Authors

In this review article, the author discusses the treatment options for patients with glioblastoma (GBM) with a special focus on the development of various oncolytic virus therapies for this condition. While the review in its present form briefly touches on a number of clinical trials that were conducted with oncolytic viruses in GBM patients in the past, it could further benefit from a better systematization of the data gathered from these trials and a more in-depth discussion of their respective outcomes. Importantly, the author appears to use chunks of texts that are seemingly taken verbatim from a number of previous publications rather than providing his own critical appraisal on these reports. This practice could convey the false impression that the author was involved in the studies he is citing, which shouldn’t be the case. For instance, the text from lines 87-89, which reads "We developed a computational approach to extract the core tumor cell intrinsic biological states of individual GBM cells from GBM single-cell RNA-sequencing (scRNA-seq) data [31,32,33]” could suggest that the author was directly involved in this study. Similarly, the text from lines 384-388: "In this work, we showed for the first time that arming VV with a scFv against TIGIT dramatically improved the parental VV's antitumor effectiveness by altering the TME's immunological state. For the first time, we also showed that the antitumor effectiveness of VV equipped with the scFv against TIGIT was further increased by the additional combination of PD-1 or LAG-3 inhibition.” Or the text that appears in lines 454-457: “We postulated that SIRT3 can function as a crucial gatekeeper for the balance between glycolytic and mitochondrial oxidative metabolism in the control of energy production for viral replication because of its critical involvement in cell metabolism.” Therefore, the author needs to rewrite all these paragraphs and address the other similar instances throughout the text by providing his own critical assessment on these topics instead. The other comments I have are:

1.     What is the current institutional affiliation of the author? He does not appear to be currently listed as a faculty member with the Department of Neurosurgery at University of Florida, Gainesville. Moreover, the contact information provided by the author is rather generic and does not suggest an ongoing affiliation with University of Florida.  

2.     In addition to the general comments above, the present review could benefit from a more detailed assessment of the outcomes from various clinical trials conducted with oncolytic therapies in GBM over the years. For instance, pertaining the poliovirus therapy conducted at Duke University, the intratumoral infusion of PVSRIPO showed for the 61 poliovirus patients enrolled in the phase I study, that the median overall survival was 12.5 months, compared to 11.3 months for the historical control group. Details like this are important and need to be discussed in a review like this for the benefit of potential readers. 

3.     The author writes in the abstract that the methodology used for his review was based on PRISMA guidelines. The PRISMA abbreviation needs to be defined accordingly.  

Author Response

thank you for the review comments.

"We developed a computational approach to extract the core tumor cell-intrinsic biological states of individual GBM cells from GBM single-cell RNA-sequencing (scRNA-seq) data"

has been correctly changed to

"We reviewed the developed computational approach to extract the core tumor cell-intrinsic biological states of individual GBM cells from GBM single-cell RNA-sequencing (scRNA-seq) data "

"In this work, we showed for the first time that arming VV with a scFv against TIGIT dramatically improved the parental VV's antitumor effectiveness by altering the TME's immunological state. For the first time, we also showed that the antitumor effectiveness of VV equipped with the scFv against TIGIT was further increased by the additional combination of PD-1 or LAG-3 inhibition.”

was changed to 

"In this work, the authors showed for the first time that arming VV with a scFv against TIGIT dramatically improved the parental VV's antitumor effectiveness by altering the TME's immunological state. For the first time, it was shown that the antitumor effectiveness of VV equipped with the scFv against TIGIT was further increased by the additional combination of PD-1 or LAG-3 inhibition."

“We postulated that SIRT3 can function as a crucial gatekeeper for the balance between glycolytic and mitochondrial oxidative metabolism in the control of energy production for viral replication because of its critical involvement in cell metabolism.”

was changed to 

"The authors postulated that SIRT3 can function as a crucial gatekeeper for the balance between glycolytic and mitochondrial oxidative metabolism in the control of energy production for viral replication because of its critical involvement in cell metabolism "

  1. I do remote research with someone at the University Of Florida. They asked me to use that affiliation when submitting for publication and I have been publishing with them on multiple articles with this affiliation. I have had any concerns regarding that. I can use either of my affiliations. Let me know.
  2. has been added as per your suggestion "A study carried out in this patient group, assessing intratumoral convection-enhanced transport of the recombinant nonpathogenic polio-rhinovirus chimera (PVSRIPO) found that all 61 patients who got PVSRIPO had a median overall survival of 12.5 months, longer than the historical control group's 11.3 months "
  3. methodology has been updated "The study was based on the PRISMA guidelines. Systematic retrieval of information was performed on PubMed. 307 articles were found in a search on oncolytic viral therapies for Glioblastoma. Out of these 83 articles were Meta-analyses, Randomized Controlled Trials, Reviews, and Systematic Reviews. 42 articles were from the years 2018 to 2023. Appropriate studies were isolated, and important information from each of them was understood and entered into a database from which the information was used in this article"

Round 2

Reviewer 2 Report (New Reviewer)

Comments and Suggestions for Authors

Accept this review in present form

Author Response

Thank you for the review comments. 

Confirming no changes required by them. 

Thanking you. 

Reviewer 3 Report (New Reviewer)

Comments and Suggestions for Authors

I thank the author for his responses. 

The author's institutional affiliations are still not clear to me. The author states that he does remote research for University of Florida. I am not entirely sure what that means. If the author is not officially affiliated with this institution, then he should list his current institutional affiliation(s) instead in the manuscript rather than making statements that cannot be verified. The author also states that he had published before with faculty members from the Neurosurgery Department at University of Florida. While that may be the case, I couldn't find any such publications. I admit that my search was not comprehensive due to time constraints. Nonetheless, I think it is important for the author to clarify his institutional affiliations and use the most current one(s). 

Author Response

Thank you for your response. 

Here is the list of publications. 

Shah, S., & Tereda, A. (2023). A comparative study of microsurgery and radiosurgery for acoustic neuroma: Tumor control, hearing preservation, and functional outcomes. Medical Imaging Process & Technology, Vol 6,(Issue 1, 2023). https://doi.org/10.24294/mipt.v6i1.2791

10/2023 Ruchika, F., Shah, S., Delawan, M., Durga, N., & Lucke-Wold, B. (2023). Cytokines and subarachnoid hemorrhage. In vitro diagnosis, 1(1), 55.

 https://doi.org/10.59400/ivd.v1i1.5 PMID: 37982005; PMCID: PMC10657139.

10/2023 Shah, S., & Lucke-Wold, B. (2023, October 10). Pharmaceutical management of hemorrhagic stroke: Optimizing outcomes following intracranial hemorrhage evacuation.Medical Imaging Process & Technology, 6(1). https://doi.org/10.24294/mipt.v6i1.2276

04/2023 Ruchika F, Neupane D, Shah S, Delawan M, Lucke-Wold B, et al. Advancing Analytics of EEG Signals. Med Discoveries. 2023; 2(4): 1031. PMCID: PMC10208433

  NIHMSID:NIHMS1890595PMID: 37228899 

Ruchika F, Shah S, Neupane D, Vijay R, Mehkri Y, Lucke-Wold B. Molecular

  Progression of Chronic Traumatic Encephalopathy in Traumatic Brain Injury, Aging and Neurodegenerative Disease. Int J Mol Sci. 2023 Jan 17;24(3):1847.Doi:10.3390/ijms24031847. PMID: 36768171; PMCID: PMC9915198.

Let me know. Thanking you. 

This manuscript is a resubmission of an earlier submission. The following is a list of the peer review reports and author responses from that submission.

Round 1

Reviewer 1 Report

Comments and Suggestions for Authors

In the review article “ Novel Therapies in Glioblastoma Treatment: Review On Glioblastoma, Current Treatment Options, Novel Oncolytic Viral And Nanoparticle Therapies” the author made an overview of the novels therapeutic treatments currently in clinical trials,  with a special focus on virotherapy based on oncolytic viruses. This is a very topical subject and deserves attention, since it is known that no therapy currently exist to treat this nefastous brain cancer . Oncolytic viruses allow to potentially hinder cancer by acting directly on cancer cells and boosting the anti-cancer immune response. The description of main viral-based therapies is quite comprehensive. However, the review in the actual form is not suitable for publication. More precisely, I have the following concerns:

1. The overview of glioblastoma (chapter 2) must be further developed with a special focus on the role of immune response in glioblastoma treatment and resistance. The paragraph on current treatment options (2.5) must be more detailed and describe available targeted therapies and immunotherapies. The authour should also better differentiate between the care of new onset glioblastoma and treatment of relapse.

2. Chapter 3 on oncolytic viral therapy is quite extensive. However, functions and mechanisms of action of the different virus must be better described. Benefits and drawbacks of the different approaches must be discussed.

3. The part on nanoparticles is not enough developed and behind the scope of this review, in my opinion. The author should remove this part and rather focus on immune and viral therapy. The title should be accordingly modified.

Specific comments

1. The bibliography is not sufficiently up- to- date. Latest literature should be added. As an example, a recent clinical study showed that associating oncolytic virotherapy and pembrolizumab, a check-point inhibitor, was safe and had some clinical benefit (Nassiri F et al Nature Medicine vpl 29 2023 et al 2023).

2. Paragraph 2.2: molecular description. The authors did not mention the classification of GB based on transcriptomic signature (classical, proneural and mesenchymal), according to Verhaak  RGW and co-workers , which is a reference in the field.  I suggest to add these bibliographic references.

3. Paragraph 2.5: please add bibliographic reference for Stupp’s gold standard reference protocol of glioblastoma treatment.

4. Figure 1: a legend fully describing the figure is missing.

Author Response

  1. has been added
  2. has been discussed in brief
  3. has been removed
  4. added
  5. added a note on it
  6. added ref
  7. added legend

Reviewer 2 Report

Comments and Suggestions for Authors

Dear Authors,

Congratulations on your work. There are major issues that should be resolved in order to make this manuscript publishable: 

1. Lack of Discussion on Recent Advances in Other Therapies: One of the most notable shortcomings of this manuscript is the absence of a comprehensive discussion regarding recent advances in other therapies related to the topic. Medical research is a rapidly evolving field, and it is crucial to provide readers with a clear understanding of how the proposed therapy or approach compares to existing methods. Failing to acknowledge recent developments in the field may make the manuscript seem outdated and less relevant.

2. Absence of Conclusions Section: A critical component of any research manuscript is the conclusions section, which should summarize the key findings and their implications. The lack of a conclusions section in this manuscript leaves readers without a clear understanding of the main takeaways and significance of the research. The author should consider adding a dedicated conclusions section to provide closure and insights into the practical implications of the study.

3. Excessive Subheadings: The manuscript contains an excessive number of subheadings, which can disrupt the flow of the text and make it challenging for readers to follow the author's arguments and ideas. It is advisable to streamline the subheadings and group related content together, creating a more coherent narrative structure. This will enhance the readability and overall organization of the manuscript.

4. Limited Discussion of Methodology: While the manuscript discusses the proposed therapy or approach in detail, it lacks a thorough discussion of the methodology used in the research. Readers should have a clear understanding of how data were collected, analyzed, and interpreted. The author should consider providing more insight into the research methods to enhance the manuscript's credibility.

5. Inadequate Citation of Recent Literature: The manuscript appears to rely heavily on older references, with limited citation of recent research publications. It is essential to incorporate recent studies and findings to demonstrate the manuscript's relevance and awareness of the current state of the field. The author should update the reference list with recent and pertinent sources.

6. Clarity and Language: The manuscript occasionally suffers from clarity issues and could benefit from improved language use. Some sentences are convoluted, and jargon or technical terms are not always adequately explained. The author should strive for greater clarity and accessibility to ensure that the manuscript is understandable to a broad readership.

7. Grammar and Proofreading: There are several instances of grammatical errors and typos throughout the manuscript. It is essential to thoroughly proofread the document to eliminate these issues and ensure a polished final product.

Author Response

  1. added discussion and conclusion
  2. added
  3. subheadings are needed to divide info systematically hence, not removed.
  4. added more info
  5. added
  6. done
  7. done

Round 2

Reviewer 1 Report

Comments and Suggestions for Authors

I thank the author for taking into account some of the comments and trying to modify accordingly the manuscript. Notably, the author added some up-to-date references,  detailed further the molecular description of glioblastoma, included a new paragragh 2.6 about the “Role of Immunosuppressive Mechanism in Glioblastoma and Resistance to Immunotherapy” and added a new Table 1 for “Treatments for new-onset glioblastoma and recurrent glioblastoma”. However, probably due to lack of time, the manuscript still does not join the level required for publication. In general, even if conclusion has been added, the whole manuscript lacks of clarity. Behind the exhaustive list of oncolytic virus still the functions and mechanisms of action of the different virus are not described. Indeed,the chapter 3 was not modified accordingly. Benefits and drawbacks of the different approaches were not described compared to current treatments, as requested. In particular, the new table 1  is difficult to read and not well organized.  I invite the author to modify accordingly the review and make a new submission. I ask also the author for next time to kindly answer point by point to each question in detail.

Author Response

  1. Conclusion and discussion have been made for clear in terms of the information available in current literature and future directions of oncolytic viruses.
  2. More information on each virus has been added to explain its mechanism and use
  3. a new table discussing advantages of each virus in point format for easy review by reader/reviewer.
  4. Table 1 has been organised and made clear by removing unnecessary information.

Reviewer 2 Report

Comments and Suggestions for Authors

Dear Authors,

Unfortunately I could not identify any changes of your mqnuscript. The response letter shoul clearely give a point by point answer to my commentaries. I consider this manuscript not acceptable foe publication.

Author Response

  1. Lack of Discussion on Recent Advances in Other Therapies: One of the most notable shortcomings of this manuscript is the absence of a comprehensive discussion regarding recent advances in other therapies related to the topic. Medical research is a rapidly evolving field, and it is crucial to provide readers with a clear understanding of how the proposed therapy or approach compares to existing methods. Failing to acknowledge recent developments in the field may make the manuscript seem outdated and less relevant.

A comprehensive discussion and conclusion have been added to talk more about certain topics related to ocolytic viruses, the future directions and other considerations

  1. Absence of Conclusions Section: A critical component of any research manuscript is the conclusions section, which should summarize the key findings and their implications. The lack of a conclusions section in this manuscript leaves readers without a clear understanding of the main takeaways and significance of the research. The author should consider adding a dedicated conclusions section to provide closure and insights into the practical implications of the study.

Conclusion has been added and has been made for clear in terms of the information available in current literature and future directions of oncolytic viruses.

  1. Excessive Subheadings: The manuscript contains an excessive number of subheadings, which can disrupt the flow of the text and make it challenging for readers to follow the author's arguments and ideas. It is advisable to streamline the subheadings and group related content together, creating a more coherent narrative structure. This will enhance the readability and overall organization of the manuscript.

The headings make the manuscript clear and categorize information properly in a systematic way. Streamling the subheads as suggested by the reviewer will make the article lose its structure and the reader might feel lost while reading it. I have not removed the subheading since I feel its necessary for the article. I hope this doesn’t lead to rejection since the author needs to have preference of writing the article in terms of its flow.

  1. Limited Discussion of Methodology: While the manuscript discusses the proposed therapy or approach in detail, it lacks a thorough discussion of the methodology used in the research. Readers should have a clear understanding of how data were collected, analyzed, and interpreted. The author should consider providing more insight into the research methods to enhance the manuscript's credibility.

“The study was based on the PRISMA guidelines. Systematic retrieval of information was performed on PubMed. Appropriate studies were isolated, important information from each of them was understood and entered into a database from which the information was used in this article.” This was exactly what was followed and has been mentioned in the article

  1. Inadequate Citation of Recent Literature: The manuscript appears to rely heavily on older references, with limited citation of recent research publications. It is essential to incorporate recent studies and findings to demonstrate the manuscript's relevance and awareness of the current state of the field. The author should update the reference list with recent and pertinent sources.

Have been added as attested by reviewer 1 above

  1. Clarity and Language: The manuscript occasionally suffers from clarity issues and could benefit from improved language use. Some sentences are convoluted, and jargon or technical terms are not always adequately explained. The author should strive for greater clarity and accessibility to ensure that the manuscript is understandable to a broad readership.

Conclusion and discussion have been made for clear in terms of the information available in current literature and future directions of oncolytic viruses. More information available for oncolytic viruses have been added in each section of the viruses.

  1. Grammar and Proofreading: There are several instances of grammatical errors and typos throughout the manuscript. It is essential to thoroughly proofread the document to eliminate these issues and ensure a polished final product.

As mentioned by both reviewers in the rating system. The quality of English and understanding ability was rated excellent. I hope that clears itself.